# Chemical Structures and Pharmacological Profiles of Ginseng Saponins

**DOI:** 10.3390/molecules24132443

**Published:** 2019-07-03

**Authors:** Ze-Yu Shi, Jin-Zhang Zeng, Alice Sze Tsai Wong

**Affiliations:** 1School of Biological Sciences, University of Hong Kong, Pok Fu Lam Road, Hong Kong SAR, China; 2State Key Laboratory of Cellular Stress Biology and Fujian Provincial Key Laboratory of Innovative Drug Target Research, School of Pharmaceutical Sciences, Xiamen University, Xiamen 361005, China

**Keywords:** ginseng saponins, chemical structure, pharmacological action

## Abstract

Ginseng is a group of cosmopolitan plants with more than a dozen species belonging to the genus *Panax* in the family *Araliaceae* that has a long history of use in traditional Chinese medicine (TCM). Among the bioactive constituents extracted from ginseng, ginseng saponins are a group of natural steroid glycosides and triterpene saponins found exclusively throughout the plant. Studies have shown that these ginseng saponins play a significant role in exerting multiple therapeutic effects. This review covers their chemical structure and classification, as well as their pharmacological activities, including their regulatory effects on immunomodulation, their anticancer effects, and their functions in the central nervous and cardiovascular systems. The general benefits of ginseng saponins for boosting physical vitality and improving quality of life are also discussed. The review concludes with fruitful directions for future research in the use of ginseng saponins as effective therapeutic agents.

## 1. Introduction

Archaeological evidence indicates that the pharmacological use of plants originated in the Paleolithic Age some 60,000 years ago. In the ancient Orient, the use of herbal medicines can be traced back at least 5000 years. Legendary Emperor, Shennong, classified hundreds of medicinal and poisonous herbs in *Shennong’s Herbal Classic* (The Classic of Herbal Medicine), the oldest pharmacopoeia in the world [1]. Ever since, humankind has been using a variety of plants as nutrients, beverages, cosmetics, dyes, and medicines to maintain health status and improve a good quality of life.

Over the centuries, botanical medicines have been widely used to treat numerous medical conditions owing to their perceived effectiveness, fewer side effects and relatively low cost. To date, traditional Chinese medicines (TCMs) are currently attracting attention worldwide as alternative and supplemental medicines [2], with ginseng, as the ‘King of all herbs’, receiving particular attention. Especially in East Asia, ginseng is deemed to be one of the most precious plants in herbal medicine.

Ginseng is a group of cosmopolitan plants with more than a dozen species belonging to the genus *Panax* in the family *Araliaceae*. Etymologically, the word *Panax* is derived from the Greek word ‘*panacea*’, meaning ‘all-healing’ and ‘longevity’. For thousands of years, ginseng has been used as a restorative of body vitality and as an adaptogen to relieve mental and physical stress and fatigue [3]. A detailed description of the medical applications of ginseng can be found in the aforementioned *Shennong’s Herbal Classic*, compiled around 200 AD. In summary, ginseng is one of the world’s most popular and nourishing medicinal herbs, and can be consumed for a long period of time without causing harm. Among the 17 different species assigned to the genus *Panax*, *Panax* ginseng (*P*. ginseng), *Panax* notoginseng (*P*. notoginseng), and *Panax* quinquefolius (*P*. quinquefolius) are the most frequently used as medicines owing to the presence within them of ginseng saponins, which have been shown to have multiple pharmacological efficacies [4].

Ginseng has become one of the most widely consumed herbal nutritional products and alternative/complementary medicine worldwide, with a total production volume of over 80,000 tons and total sales of approximately $2.1 billion in 2013. Global demand for ginseng products, at the end of 2016 was an estimated to total 5.2 million kilograms, and the global ginseng market is also predicted to be worth some $7.51 billion by 2025 [5].

Ginseng contains various bioactive monomers, including ginseng saponins, fatty acids, polysaccharides, and mineral oils. Among its diverse constituents of ginseng, its pharmacological activities can be attributed primarily to the triterpene saponins, generally known as ginsenosides [6]. Up to now, over 100 ginseng saponins have been identified from the 11 different *Panax* species since their first description in the 1960s by Shibata’s group [7]. Ginseng saponins were first isolated by thin layer chromatography (TLC) and named alphabetically according to how far they migrated on the plate in a given system, e.g., Rb1, Rb2, Re, Rc, and so on [8]. 

Recent years have seen renewed interest in investigating the pharmacological benefits of ginseng in treating mammalian diseases. This review first describes the chemical structure of common ginseng saponins and their differing carbohydrate moieties, and then summarizes the therapeutic properties of the key saponins in terms of boosting physical vitality, as well as their regulatory effects on immunomodulation, their anticancer effects, and their functions in the central nervous and cardiovascular systems.

## 2. Chemical Structure and Classification of Ginseng Saponins

The basic structure of ginseng saponins comprises a hydrophobic, steroidal four-ring system [9] with a trans relationship and diverse carbohydrate moieties (i.e., glucose [glc], rhamnose [rha], xylose [xyl], and arabinose [ara]) attached to the carbon-3 (C3), carbon-6 (C6), and carbon-20 (C20) positions [10]. Table 1 summarizes the different carbohydrate moieties of the key ginseng saponins. Each ginseng saponin can be further classified as belonging to the dammarane type, including the panaxadiol-type (PPD) group, which contains a hydrogen atom at C6 (i.e., Rb1, Rb2, Rb3, Rc, Rd, Rg3, Rh2, Rs); the panaxatriol-type (PPT) group, which contains a C6 sugar side-chain (i.e., Re, Rf, Rg1, Rg2, Rh); or the oleanane type, which has two minor classes, the oleanolic acid group (i.e., Ro) and ocotillol group (i.e., F11), based on their chemical structures [11]. Figure 1 summarizes the chemical structures of the three basic ginseng saponin types. 

However, ginseng saponins are not easily absorbed by the body through the intestines because of their hydrophilicity, although a small amount can be absorbed in the gastrointestinal tract after orally administration. Reports suggest that the intestinal flora can degrade saponins into secondary glycosides or aglycones through hydrolysis of the glycosidic bonds and regulate their pharmacokinetics after oral intake, which in turn further determines their pharmacological efficacies [12]. As shown in Figure 2, upon their oral consumption, ginseng saponins are partly transformed into PPD and PPT through a series of deglycosylation procedures via acid hydrolysis and intestinal bacterial actions [13]. All the metabolites, such as compound K (CK), are nonpolar compared to the parental components, which can easily be absorbed in the gastrointestinal tract and express biological actions.

## 3. Key Saponins Found in Ginseng

Of the more than 100 different ginseng saponins, Rb1, Rb2, Rc, Rd, Re, Rf, and Rg1 are the most abundant in raw ginseng roots. Given that they comprise more than 90% of the total saponins in *P*. ginseng [14,15], they are also the most frequently studied. In one prior study, the crude saponin measured in 4-year-old dried ginseng accounted for over 2% of the entire plant, and the Rb1, Rf, and Rg1 content in 30 piece-graded curved ginseng samples was 2.2–4.7, 0.4–1.3, and 1.6–4.0 mg/g, respectively [16]. 

Of the two major ginseng saponin groups, the dammarane type dominates the oleanane type in both amount and structural varieties. The dammarane type includes Rb1 and Rg1 whose genuine aglycones, are protopanaxadiol (PPD) and protopanaxatriol (PPT), respectively, whereas the oleanane type is represented by only one minor saponin, i.e., Ro, whose aglycone is oleanolic acid.

According to a DESI-tandem mass spectrometry-based study, Rg1 and Rf are highly concentrated within the outer bark and inner core areas of the ginseng root, whereas Rd, Re, and Ra1 are distributed at high concentrations in the bark and at low concentrations in the center [17]. Rb1 is abundant in the roots, rhizomes, and root hairs of ginseng, compared with the stem and leaves [14].

## 4. Pharmacological Profiles and Mechanisms of Ginseng Saponins

A great number of studies have examined the various mechanisms underlying the efficacy of ginseng saponins. Some of their physiological effects may confer general health benefits. For example, there are reports of the general bolstering of the immune system, anti-inflammatory effects, anti-hepatoxicity effects, and protective actions against mammalian tumor cell lines and non-organ-specific cancers [18]. In addition, ginseng saponins have been shown to exert various pharmacological effects with newly emerging mechanisms, including cardiovascular protective activity and anti-inflammatory, antiviral, and immunoregulatory effects. The next section provides a brief introduction to these effects and activities, and Figure 3 summarizes the pharmacological profiles and related mechanisms of ginseng saponins.

Their hydrophobic steroid backbones allow ginseng saponins to intercalate into the hydrophobic interior of the lipid bilayer and further interact with the polar heads of the membrane phospholipids, which may in turn allow them to readily enter cells by simple diffusion and modulate cellular functions by binding to cytoplasmic receptors [19]. The rapid non-genomic action caused by steroids has recently been widely recognized [20]. Several mechanisms for these effects have been proposed, including changes in the membrane fluidity and activity of steroid hormones on plasma membrane receptors. Ginseng saponins may also modify the membrane protein structure by changing the membrane dynamics and modulating the activity of ion channels, membrane-bound receptors, and enzymes. Consequently, a single ginseng saponin may be capable of reflecting its pharmacological profiles via multiple mechanisms.

### 4.1. General Effects of Ginseng Saponins

The classical beneficial effects of ginseng are to replenish vital energy and increase longevity. Therefore, since ancient times, ginseng has been considered a panacea that can provide eternal youth, revitalize the body and mind, and enhance physical strength and vigor. Contemporary pharmacological studies have revealed the adaptogenic activities of ginseng and its bioactive molecules acting against stress and fatigue, as well as its non-specific resistance to various pathological factors in helping to rejuvenate the body. For instance, ginseng saponins have been shown to stimulate the formation of blood vessels and improve blood circulation in the brain, thereby enhancing memory and cognitive abilities [21]. 20(*R*)-Rg3 may have beneficial effects against fatigue by increasing the residence time of Rg3 in the nasal cavity and enhancing the absorption through the nasal mucosa [22]. Moreover, PPD (10 mg/kg) and PPT (5 and 10 mg/kg) administration has been reported to exert anti-fatigue and anti-stress effects in mice model [23]. 

### 4.2. Effects on the Central Nervous System (CNS)

Ginseng saponins have been reported to boost memory and learning, and recent studies have shown them to play a role in the treatment of neurological diseases. For example, animal studies have demonstrated that Rb1, CK [24] and Rg1 can prevent scopolamine-induced memory deficits. In a C57BL/6 mice model, CK reversed memory impairment by inducing translocation of the nuclear factor erythroid 2-related factor (Nrf2), which further enhanced the cell stress response. Rb1 has also been found to increase the uptake of choline in the central cholinergic nerve endings [25], which have been implicated in mediating memory and learning.

The use of a combination use of ginseng saponins was associated with a prolonged drug induced sleeping time in a mouse model [26], suggesting a central nervous system (CNS)-depressing property caused by the mixture. Other studies have indicated that ginseng saponins also have neuroprotective effects. In one study, for example, Rb1 treatment inhibited GSK-3*β*-mediated C/EBP homologous protein (CHOP) signaling, thereby preventing neuronal apoptosis due to endoplasmic reticulum stress [27]. In another, Rg3 pretreatment attenuated Hcy-induced neurotoxicity by reducing the intracellular Ca^2+^ elevation via N-methyl-D-aspartate receptor (NMDAR) activation [28]. Rg3 was also demonstrated to protect the vascular endothelial cells via the estrogen receptor (ER), which is similar to the effect exerted by 17-*β*-estradiol. Furthermore, Rg1 also exerts an estrogenic effect by activating extracellular regulated protein kinases (ERK) and Akt signaling in an ER*α* phosphorylation-dependent manner, which results in memory improvement [29]. Hence, ginseng saponins may play a CNS modulatory function by interacting with hormone receptors.

Several common pharmaceuticals and supplements with antioxidant properties have been investigated as therapeutic agents for various diseases. Ginseng saponins have been proven to exert a CNS protection effect that is attributed to their antioxidant properties, which operate by increasing internal antioxidant enzymes. For example, Rb1 exerts antioxidant effects in the ischemic hippocampal neurons [30]. 

### 4.3. Anticarcinogenic Activities

The significant antitumor properties of ginseng saponins are known to be the result of their anti-inflammatory, anti-proliferation, anti-metastasis, and anti-angiogenesis effects, and reversal of multi-drug resistance. In addition, their low toxicities and few side effects make them a promising prospect for anticancer research. Ginseng saponins exert anticarcinogenic effects both in vitro and in vivo through various mechanisms, including direct cytotoxicity, growth suppression, differentiation induction, and metastasis inhibition.

For example, Rh2 has been shown to suppress tumor cell growth in breast cancer [31], prostate cancer, and leukemia [32], and to arrest melanoma cell cycle progression in the G1 phase. Orally administered and subcutaneously injected Rh2 was reported to inhibit the growth of human ovarian cancer cells transplanted into nude mice, and significantly prolong the overall survival time. Rg3 treatment in lung cancer cells was found to suppress NF-κB activity to a significant degree, thereby inhibiting tumor progression [33], and 20(*S*)-Rg3 to reduce human gallbladder cell (GBC) progression to a remarkable extent by activating the p53-mediated apoptotic pathway [34]. In another study, Rg3 displayed antitumor activity in LNCaP cell line via the down-regulation of the androgen receptor (AR) and other prostate cancer biomarkers [35]. Rb1 has been shown to inhibit angiogenesis via induction of the anti-angiogenic modulator pigment epithelium-derived factor (PEDF) [36]. Similarly, Rb2 and Rg3 were found to suppress angiogenesis and metastasis both in vitro and in vivo [37].

Compound K (CK), an important active metabolite of PPD-type ginseng saponin bio-transformed by intestinal microorganisms after oral administration, is drawing increasing attention recently owing to its potent anticancer benefits [38]. Research has shown that CK inhibited angiogenesis in human umbilical vein endothelial cell (HUVEC) by suppressing sphingosine-1-phosphate (S1P) induced cell migration via sphingosine kinase-d1 (SphK1) modulation [39]. Besides, CK was found to decreases basic fibroblast growth factor (FGF)-induced angiogenesis by down-regulating the phosphorylation of p38-mitogen-activated protein kinase (p38-MAPK) and Akt. A recent finding documented by our group was that CK can reduce resistance to chemotherapeutic drugs and inhibit the epithelial-to-mesenchymal transition via the Wnt/*β*-catenin pathway in ovarian cancer stem cells (CSC), indicating that it may be a good candidate for CSC weakening [40].

Ginseng saponins have also demonstrated strong chemoprotective and chemotherapeutic properties in a wide range of experimental studies both in vitro and in vivo. It seems that a large-scale, controlled clinical study is needed to validate these results.

### 4.4. Immunomodulatory Effects

The immunomodulatory and anticarcinogenic activities of ginseng saponins are often discussed together. For example, Rg1 has been reported to increase both humoral and cell-mediated immune responses [41]. Furthermore, an enhanced antitumor immunological response contributed to preventing tumor recurrence, i.e., Rh2, when used as an adjuvant, was found to trigger CD4+/CD8a+ T-lymphocyte infiltration in tumor tissues and to increase T-lymphocyte cytotoxicity with cancer chemotherapeutic drugs [42]. CK and PPT type saponins have also been shown to promote the differentiation of peripheral blood monocytes into dendritic cells (DC), suggesting a potential use in cancer immunotherapy [43]. In addition, one in vivo study found that three days of Rg1 (10 mg/kg) administration in mice prior to immunization increased the number of spleen plaque-forming cells and antigen-reactive T-lymphocytes. The ratio of T-helper cells and the natural killer activity of splenocytes were also enhanced [41]. 

Ginseng saponins can also be considered good drug candidates or supplements for transplantation and autoimmune disorders owing to their immunosuppression ability. For example, Rd and CK up-regulate regulatory T cell (Treg) differentiation in vitro by increasing the production of immunosuppressive cytokines TGF-*β* and IL-35. Moreover, it has also been reported that CK can reduce adjuvant-induced arthritis by up-regulating Treg cells and suppressing T cell activation in rats [44].

To sum up, there is growing evidence of the immunomodulatory properties of ginseng saponins. Further investigation is needed to uncover the underlying mechanism of their cancer-related immunotherapy in the tumor microenvironment.

### 4.5. Cardiovascular System

Cardiovascular disease (CVD) is the leading cause of death globally. In recent years, the effects of ginseng saponins on CVDs have been researched extensively because of their intrinsic properties of controlling reactive oxygen species (ROS) and nitric oxide (NO) production, and the ability to activate various receptors in endothelial cells. For example, Rb1 has been shown to preserve both endothelial NO production in HUVEC from the toxic effects of oxidized low-density-lipoproteins and the endothelium-dependent relaxation of porcine coronary arteries exposed to homocysteine [45]. In another study, 20(*S*)-Rg3 enhanced endothelial NO synthase (eNOS) production by rapidly inducing the ERK/Akt signaling pathway via the activation of peroxisome proliferator-activated receptor-*γ* (PPAR-*γ*) and induced HUVEC proliferation [46]. Similarly, Rb1 acutely up-regulated eNOS by phosphorylating Ser1177 and increased NO production in human aortic endothelial cells [47].

Ginseng saponins have also been documented to exert an ion channel-regulating effect, with one study reporting Rb1 to exert an antihypertrophic effect by releasing NO and to attenuate the expression of NFAT3 and GATA4 transcription factors in cardiomyocytes by reducing the calcineurin signal transduction pathway [48]. Another study proved that Rg1 treatment to directly suppress the Ca^2+^ channel to inhibit Ca^2+^ overload in cardiomyocytes and improve the production of antioxidant enzymes in cardiac myocytes during myocardial ischemia [49]. In ischemia reperfusion treatment, Re effectively activates the cardiac K^+^ channels via eNOS activation by means of an ER*α* related non-genomic pathway [50].

In addition, ginseng saponins have a protective effect on tissue damage, rendering them a novel therapeutic for heart failure. For example, Re has been shown to protect cardiomyocytes from oxidative injury by scavenging hydrogen peroxide and hydroxyl radicals [51]. Saponins from *P*. notoginseng have also been found to protect the heart against doxorubicin induced cardiotoxicity and to block monocrotaline-induced cardiac hypertrophy in rats [52].

### 4.6. Diabetes Mellitus

Owing to their effects on the endocrine system, ginseng saponins are widely used in the adjuvant treatment of diabetes and diabetic complications. For instance, Rb1 has a significant antihyperglycemic effect and increases insulin sensitivity, and is thus used clinically to treat diabetes mellitus (DM) [53]. Furthermore, Rb1 is also useful for treating obesity, which is mediated in part by reducing food intake and body weight in addition to lowering glucose [54]. The acute intraperitoneal injection of Rb1 (10 mg/kg) was demonstrated to significantly suppress food intake in rats, and chronic Rb1 treatment was found to possess no deleterious side effects in obese rats [55].

In addition, Rb1 was reported to improve insulin resistance in a high-fat diet-induced type 2 DM mouse model by reducing blood glucose [56]. Oral Re intake has also been shown to significantly decrease fasting blood glucose levels and improve glucose tolerance and systemic insulin sensitivity without affecting body weight in mice [57]. These findings indicate that ginseng saponins may be useful for improving glucose tolerance and insulin resistance in patients with type 2 diabetes.

Diabetic cardiomyopathy manifests primarily as myocardial dysfunction in the absence of other heart disease, and may eventually progress to heart failure [58]. In vitro experiments showed that Rh2 activated PPAR*δ* in cardiomyocytes cultured in high glucose, in turn by inhibiting the expression of STAT3, reducing cardiac fibrosis, and protecting against diabetic cardiomyopathy [59]. In a rat model, Rh2 improved heart function in streptozotocin-induced type 1 diabetes.

### 4.7. Clinical Application

Owing to their incomplete pharmacokinetic parameters and unknown toxicities [60], individual ginseng saponins have not yet been investigated in clinical studies. However, ginseng and ginseng extract have yet been widely used in clinical practice for thousands of years.

For example, in the context of Alzheimer’s dementia (AD), ginseng-treated patients exhibited a clinical improvement in cognitive function at 4 weeks, with continuing effects documented at 12 weeks. According to a more recent study, treatment with relatively large amounts of ginseng extract for more than 12 weeks improved the frontal assessment battery, an indicator of frontal cortical activity, in elderly patients with AD [61]. Oral administration of a ginseng saponin at 4.5g per day in AD patients had been shown to have a positive effect on the frontal lobe, which could enhance memory function [62]. Epidemiological studies of ginseng intake in 4600 cancer patients showed those who had taken ginseng to have a lower risk of developing certain cancers (i.e., stomach, liver, and lung) than those who had not. Moreover, increased intake leads to a lower danger ratio, proving ginseng’s usefulness for primary prevention [63].

## 5. Other Active Ingredients in Ginseng

In addition to saponins, ginseng also comprises several other active constituents, including polysaccharides, phenolic compounds, and alkaloids. The more biologically active carbohydrates isolated from ginseng are acidic polysaccharides, known as ginsenan, which have a typical pectin structure. Polysaccharides are the most abundant components of ginseng. This class of compounds was first isolated and documented in the 1960s [64]. To date, about 80 polysaccharides isolated from ginseng have demonstrated promising pharmacological activities, including antitumor, immunomodulation, and anti-fatigue effects. However, few studies have tested them relative to ginseng saponins, and the latter also have more significant effects. A clinical trial comparing ginseng saponins and polysaccharides revealed the saponin fraction of red ginseng to decrease the augmentation index in arterial stiffness patients, but no effects were observed for the polysaccharides [65].

More than 10 phenolic compounds have been identified in *P*. ginseng, including elemicin and dauricine, and been found to exert various biological properties, including antitumor, antioxidant, and anti-inflammatory activities. Flavonoids are a group of polyphenolic compounds that consist of two phenyl rings and a heterocyclic ring and are universally present in plants. Flavonoids are believed to have health promoting properties due to their antioxidant activities [66]. Kaempferol is the representative flavonoid in *P*. ginseng.

Alkaloids, such as fumarine and girinimbi, are a minor non-saponin component of *P*. ginseng. Three *β*-carboline alkaloids were isolated from the ginseng root for the first time in 1987 [67], and two additional *β*-carboline alkaloids were reported the following year.

## 6. Conclusion and Future Prospects

Ginseng saponins are believed to inhibit ROS production, enhance immune function and CNS function, and prevent CVD. Moreover, their ability to kill cancer cells with relatively little toxicity to normal cells makes ginseng saponins an attractive candidate for drug development. However, as with other natural compounds derived from plants, their poor water solubility, permeability, absorption, and bioavailability have restricted their application. It is, therefore, hoped that more studies directed at designing and synthetizing novel derivatives and developing suitable drug delivery systems for ginseng saponins will be carried out in future.

Natural compounds are genetically encoded by their producing organisms, which means that they are the products of natural selection directed at benefiting their producers. The advantages of those compounds are thus the result of their effective engagement with their biologically relevant targets and receptors [68]. Botanic natural compounds are well-tolerated and do not exert toxic effects even at high doses. Moreover, some natural compounds may increase the efficacy of conventional therapy by mutually reinforcing the combined therapeutic actions [40,69]. Ergo, the use of phytomedicine has been valued throughout human history. Although ginseng has been widely used for thousands of years, demonstrating various health benefits for a wide range of physiological ailments, there is surprisingly little scholarly knowledge of the molecular mechanisms of action for its bioactive ingredients. There is increasing evidence to suggest that ginseng saponins are involved in the regulation of hormone receptors, which are linked to a diverse array of signaling pathways. Thus, research identifying the cellular targets responsible for the activities of ginseng saponins and characterizing their activities are warranted, which is essential for the development of multi-target phytomedicine.

## Figures and Tables

**Figure 1 molecules-24-02443-f001:**
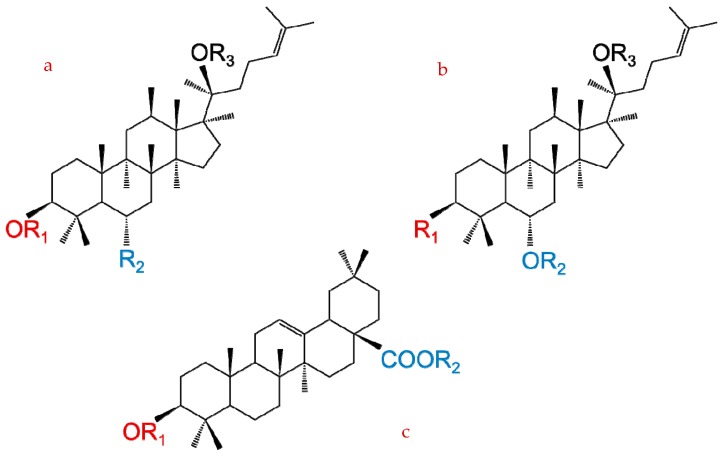
Chemical structures of three ginseng saponin types: (**a**) protopanaxadiol type (PPD); (**b**) protopanaxatriol type (PPT); and (**c**) oleanolic acid type.

**Figure 2 molecules-24-02443-f002:**
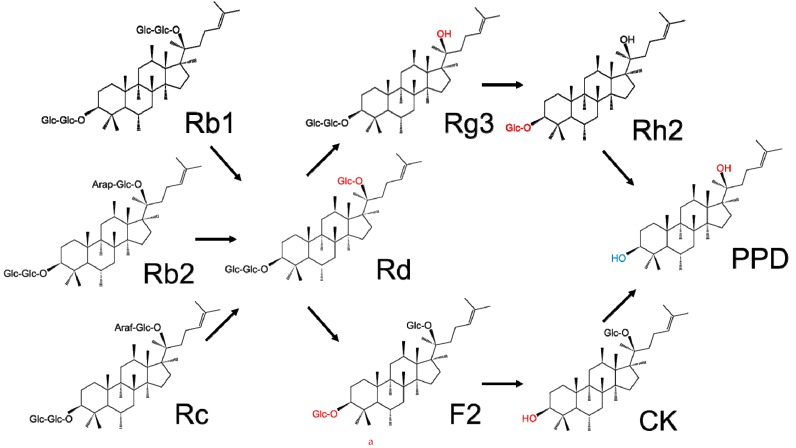
Main metabolic processes of (**a**) 20(*S*)-protopanaxadiol- and (**b**) 20(*S*)-protopanaxatriol-type ginseng saponins. The changed carbohydrate moieties are highlighted in red. For the daughter ginseng saponin comes from two different parental ginseng saponins, one of the changed carbohydrate moieties is highlighted in red and the other is highlighted in blue.

**Figure 3 molecules-24-02443-f003:**
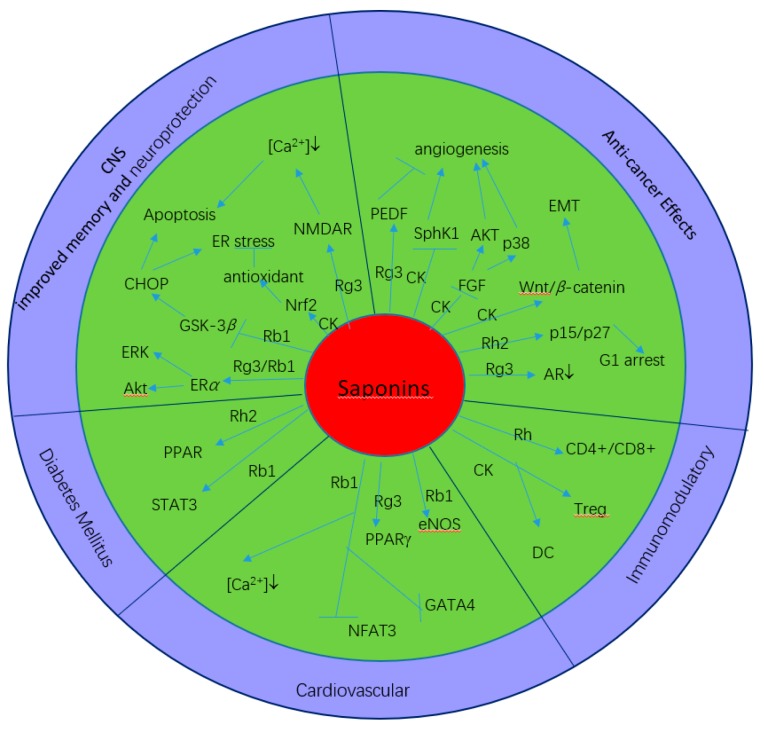
Pharmacological profiles and mechanisms of ginseng saponins.

**Table 1 molecules-24-02443-t001:** Summary of ginseng saponin carbohydrates.

	Formula	Carbohydrate Moieties		
		R1 (C3)	R2 (C6)	R3 (C20)
Protopanaxadiol	C_30_H_52_O_3_	H	H	H
Ra1	C_58_H_98_O_26_	Glc-Glc	H	Glc-Ara(p)-Xyl
Ra2	C_58_H_98_O_26_	Glc-Glc	H	Glc-Ara(f)-Xyl
Ra3	C_59_H_100_O_27_	Glc-Glc	H	Glc-Glc-Xyl
Rb1	C_54_H_92_O_23_	Glc-Glc	H	Glc-Glc
Rb2	C_54_H_90_O_22_	Glc-Glc	H	Glc-Ara(p)
Rb3	C_53_H_90_O_22_	Glc-Glc	H	Glc-Xly
Rc	C_53_H_90_O_22_	Glc-Glc	H	Glc-Ara(f)
Rd	C_48_H_82_O_18_	Glc-Glc	H	Glc
Rg3	C_42_H_72_O_13_	Glc-Glc	H	H
Rh2	C_36_H_62_O_8_	Glc	H	H
Rs1	C_36_H_92_O_23_	Glc-Glc-Ac	H	Glc-Ara(p)
Rs2	C_55_H_92_O_23_	Glc-Glc-Ac	H	Glc-Ara(f)
Rs3	C_44_H_74_O_14_	Glc-Glc-Ac	H	H
Compound K	C_36_H_62_O_8_	H	H	Glc

Protopanaxatriol	C_30_H_52_O_4_	H	H	H
Re	C_48_H_82_O_18_	H	Glc-Rha	Glc
Rf	C_42_H_72_O_14_	H	Glc-Glc	H
Rg1	C_42_H_72_O_14_	H	Glc	Glc
Rg2	C_42_H_72_O_13_	H	Glc-Rha	H
Rh1	C_36_H_62_O_9_	H	Glc	H
F1	C_36_H_62_O_9_	H	H	Glc
F3	C_41_H_70_O_13_	H	H	Glc-Ara(p)

Oleanolic acid				
Ro	C_48_H_76_O_19_	GlcUA-Glc	Glc	

Ac: acetyl; Ara(f): α-L-arabinofuranosyl; Ara(p): α-L-arabinopyranosyl; Glc: β-D-glucopyranosyl; GlcUA: β-D-gulcuronic acid; Rha: α-L-rhamnophranosyl; Xyl: β-D-xylopyranosyl

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
