# Peer review of "Chemical Structures and Pharmacological Profiles of Ginseng Saponins"

_molecules, 2019, doi:10.3390/molecules24132443_

Round 1

Reviewer 1 Report

Have only a few technical corrections (see enclosed file, please). Well done.

Author Response

1.     The corrections are made accordingly.

Reviewer 2 Report

This review by Shi and coworkers aims to present known information about ginseng saponins, a group of natural steroid glycosides and triterpene saponins that are found in the ginseng plant. These compounds have been shown to exhibit a number of biological activities and play a significant role in exerting multiple therapeutic effects. 

There have been reviews on these compounds published as recently as 4 years ago (Shin BK, Kwon SW, Park JH. Chemical diversity of ginseng saponins from Panax ginseng. J Ginseng Res. 2015;39(4):287–298. doi:10.1016/j.jgr.2014.12.005) although I do see that there are publications included in this review after this, so there is some merit in the publication of this review, provided the following changes are made: 

Line 12: Being some (not one)

Lines 31-33: Sentence is confusing and does not make sense

Lines 56-57: Sentence does not make sense and the terms dammarane and oleanane should be clarified, particularly as they are used later on (Section 3) 

Line 70: four trans ring? 

Table 1 caption: carbohydrates

Figure 1: The quality of the structures needs to be improved

Figure 2 needs to be remade and significantly improved - the compound names need to be clearer and the structures themselves should be larger. Furthermore, the structures should be formatted more clearer - it is very difficult to follow and see how the structures relate to each other. The "S"'s in the figure caption should be italicised. Also the text describing this Figure should be improved and expanded. 

In addition to the above changes, the manuscript does require extensive english editing before publication. 

Author Response

1.     Line 13: ‘Being some of’ has been changed into ‘Among’.

1.     Lines 32-33: The sentence is revised to make it clear.

2.     Lines 56-59: The sentence is revised to make it clear. The terms dammarane and oleanane are defined in lines 74-79 of section 2 on Chemical Structure and Classification of Ginseng Saponins.

3.     Line 70, the word ‘trans’ has been set in italics to make it proper (Ref.#9).

4.     Table 1 caption: ‘carbohydrate’ has been changed into ‘carbohydrates’.

5.     Figure 1: The structures are remade.

6.     Figure 2: the structures are remade, and the caption is revised as suggested.

7.     In rewriting the revised manuscript, we have also tried to improve the English and the grammar by professional English editing.

Reviewer 3 Report

It is an interesting review on the pharmacology and physicochemistry of Ginseng Saponins. The overall quality of the paper regarding readability and language is very good.To my opinion the paper is useful for the readers and I support its publication in the present form.

Author Response

N.A.

Reviewer 4 Report

This is about the manuscript no 528703, a very good review about the ginseng saponins.

The following aspects must be highlighted:

- page 2 (lines 48-62) contains a few data from 2012-2013; they give the impression that the manuscript was written in that period;

- the description of their effects against stress and fatigue is too short described in part 4.1;

- abbreviation HUVEC is not explained at its first use on page 9;

- thier application in Alzheimer’s dementia is very short described;

- just 21 from 101 references are new (from the last 5 years)

Author Response

1.     Lines 52-53: the ginseng market data of 2016 are added. Line 57, ‘Until the year 2012’ has been changed into ‘Up to now’, and ‘approximately’ has been changed into ‘over’.

2.     Lines 153-156, two references (Ref.#21 and 22) regarding anti-fatigue/anti-stress effects of ginseng saponins have been added; one reference regarding memory enhancing has been removed.

3.     Line 206, the full name of HUVEC, human umbilical vein endothelial cell has been added as its first use.

4.     Lines 292-296, two recent clinical trials in elderly AD patients has been added (Ref.#60 and 61).

5.     Ref.#25 and 30 have been replaced. 37 earlier references have been removed (Ref.#3, 7, 12, 14, 16, 18, 22, 24, 28, 29, 33-35, 37, 39, 44, 46, 48, 50, 51, 59, 61, 62, 74, 83, 84, 86, 87, 90-92, 95, 96, 98, 100, 101, 103). Now there are 32 references published within the recent 5 years.

Round 2

Reviewer 2 Report

This is a review of a resubmission of an article that I have seen previously. While I can see that the authors made an effort to address all of my concerns, I think there are still a couple of improvements to be made, including: 

Line 70: change it to "four-ring system with a trans relationship" would be far clearer, not just italicising the trans

Figure 2: While better than in the first submission, this figure is still not well done. The structures should be bigger (and perhaps arrows a little smaller). 

Overall, if these changes were made, I would recommend acceptance of the article into Molecules.